# Insights Gained from RNA Editing Targeted by the CRISPR-Cas13 Family

**DOI:** 10.3390/ijms231911400

**Published:** 2022-09-27

**Authors:** Li Liu, De-Sheng Pei

**Affiliations:** 1Chongqing Institute of Green and Intelligent Technology, Chongqing School of University of Chinese Academy of Sciences, Chinese Academy of Sciences, Chongqing 400714, China; 2University of Chinese Academy of Sciences, Beijing 100049, China; 3School of Public Health and Management, Chongqing Medical University, Chongqing 400016, China

**Keywords:** CRISPR/Cas13, Cas13d, Cas13X, RNA cleavage activity, CRISPR-Cas VI system

## Abstract

Clustered regularly interspaced short palindromic repeat (CRISPR)/CRISPR-associated protein (Cas) systems, especially type II (Cas9) systems, have been widely developed for DNA targeting and formed a set of mature precision gene-editing systems. However, the basic research and application of the CRISPR-Cas system in RNA is still in its early stages. Recently, the discovery of the CRISPR-Cas13 type VI system has provided the possibility for the expansion of RNA targeting technology, which has broad application prospects. Most type VI Cas13 effectors have dinuclease activity that catalyzes pre-crRNA into mature crRNA and produces strong RNA cleavage activity. Cas13 can specifically recognize targeted RNA fragments to activate the Cas13/crRNA complex for collateral cleavage activity. To date, the Cas13X protein is the smallest effector of the Cas13 family, with 775 amino acids, which is a promising platform for RNA targeting due to its lack of protospacer flanking sequence (PFS) restrictions, ease of packaging, and absence of permanent damage. This study highlighted the latest progress in RNA editing targeted by the CRISPR-Cas13 family, and discussed the application of Cas13 in basic research, nucleic acid diagnosis, nucleic acid tracking, and genetic disease treatment. Furthermore, we clarified the structure of the Cas13 protein family and their molecular mechanism, and proposed a future vision of RNA editing targeted by the CRISPR-Cas13 family.

## 1. Introduction

The structure of the CRISPR-Cas system has gone through more than 20 years from discovery to function revelation, and its basic structure and molecular mechanism have been gradually elucidated in recent years. In 1987, a Japanese scientist stumbled upon a 29 bp repeating sequence while studying the isozyme conversion of alkaline phosphatase in *Escherichia coli*. Unfortunately, there is not enough DNA sequence data available, and Ishino et al. could not explain the function of these unique sequences [1]. Mojica et al. first identified a 30 bp DNA fragment that repeated at a regular distance in the genome of *Haloferax mediterranei*, calling this element “short regularly spaced repeats” (SRSRs) in 1993 [2]. In 2002, these SRSRs were named “CRISPR” by Mojica [3]. Three years later, several bioinformatics teams almost simultaneously found that CRISPR spacer sequences were consistent with foreign DNA sequences of invading bacteria, suggesting that the CRISPR system might play a major role in microbial immunity [4,5,6]. In 2011, the CRISPR-Cas system was first classified according to the DNA structure and the dynamic evolution of the CRISPR-Cas system [7]. In 2012, Jinek et al. elucidated the mechanism of CRISPR/Cas9 and highlighted the potential use of this system for programmable genome editing [8]. The next year, Mali et al. successfully applied the technique to the human genome and other organisms [9]. Two years later, two studies reported type II CRISPR systems from *Streptococcus thermophilus* and *Streptococcus pyogenes* and successfully edited the genomes of mammalian cells [10]. In 2015, the discovery of Cpf1 (Cas12a) provided a more flexible way of selecting DNA targets compared to Cas9. Cpf1 (Cas12a) recognizes AT-rich PAMs, and it can target AT-rich regions. However, Cpf1 (Cas12a) has difficulty finding targets in GC-rich regions [11]. Therefore, Cpf1 (Cas12a), together with Cas9, expands the target sites for selection. Cas13, a type VI CRISPR system-related protein discovered in 2016, is an RNA-guided and RNA-targeted ribonuclease protein family, which avoids permanent damage to the DNA of organisms [12,13,14]. Since then, due to the emergence of RNA editing targeted by the CRISPR-Cas13 family, CRISPR technology has broadened its research scope and ushered in a bright prospect for the future.

The CRISPR-Cas systems are an adaptive immune system derived from prokaryotes, which widely exist in 40% of bacteria and 90% of archaea to protect bacteria and archaea from exogenous genetic invasion through the interaction of RNA and CRISPR-related proteins [15,16,17]. The CRISPR-Cas system consists of three parts: the leader sequence (LS), an operon containing a set of *cas* genes, and a CRISPR DNA array. The leading sequence is about 200–500 bp upstream of the first CRISPR repeat sequence rich in A/T and provides recognition sites [18,19]. The CRISPR DNA array is composed of 21–48 bp length repeats and spacer sequences, and a hairpin structure can be formed and repeated up to 250 times, which is the binding region of Cas protein [20]. Spacers are separated by repeated regulatory sequences and form the CRISPR array with leader sequences and repeated sequences [5,6]. The CRISPR-associated gene is highly conserved near the CRISPR locus. The Cas protein encoded by the CRISPR-associated gene contains endonucleases, helicases, and a binding domain with ribonucleic acid, which can recognize foreign DNA and cut the invading DNA through site-specific cleavage [21].

Bacteria and archaea use the adaptive immune system to protect exogenous genetic components from phages and nucleic acids. CRISPR-mediated adaptive immunity consists of three phases: adaptation, maturation, and interference (Figure 1) [21]. In the adaptation phase, the CRISPR-Cas system acquires new spacer sequences. When the small gene fragments from the invader enter bacteria and archaea containing the CRISPR-Cas system, the host’s CRISPR-associated protein complex binds to the protospacer adjacent motifs (PAM) of the gene fragments. A new spacer sequence was formed under the action of related proteins and established structural motifs for the acquired immunity of bacteria [22]. Mature CRISPR RNA (crRNA) was generated by the CRISPR-Cas expression system during the maturation phases. When foreign nucleic acids invaded again, the CRISPR sequence of the CRISPR-Cas system was transcribed into a sequence-specific CRISPR RNA precursor (pre-crRNA), which was further processed and spliced by Cas proteins to form mature CRISPR RNA (crRNA). Interference phases: ribonucleoprotein complex with crRNA and Cas protein can detect and bind the target invading genome (DNA/RNA) by crRNA, and eventually leads to the breakage or degradation of the invader genome (DNA/RNA) [23,24].

This study mainly reviews the systematic classification, defense mechanism, and the application prospect of RNA editing targeted by the CRISPR-Cas13 family, and especially highlights the latest research progress of the CRISPR-Cas13 family, including Cas13d and Cas13X. The existing problems of RNA editing systems based on the Cas13 family are also analyzed.

## 2. Classification and Function of Type VI CRISPR Systems

### 2.1. Type VI CRISPR-Cas Family

The CRISPR system is classified mainly according to its evolutionary development of the CRISPR system, Cas protein sequence, and genomic locus architecture [7]. The adaptation module in several CRISPR-Cas systems possesses endonuclease Cas1 and structural subunit Cas2, which are critical for acquiring spacers [25]. According to the structural composition of the Cas effector protein complex, the currently known CRISPR-Cas system can be divided into two classes (class 1 and class 2), and each can be further divided into three types. Among them, the class 1 system (including type I, III, and IV) exist in bacteria and archaea, requiring complex multiple subunits and crRNA to form the CRISPR-Cas complex. Type I and type III effectors have a common origin [26,27]. In contrast, the class 2 system (including type II, V, and VI) exists almost exclusively in bacteria, requiring a single Cas protein and a crRNA to form the CRISPR-Cas complex [13,26,28,29]. In the class 2 system, Cas9 is a type II effector complex containing two unrelated nuclease domains (HNH and RuvC), while the type V effector Cpf1 (Cas12a) contains only one nuclease domain (RuvC-like). Of note, type VI CRISPR systems including the CRISPR-Cas13 family are unique because their Cas proteins contain two conserved “higher eukaryotic and prokaryotic nucleotide binding (HEPN)” domains with RNase activity [30,31,32,33]. The property difference between Cas9 and Cas13 can be seen in Table 1. Due to the complexity of multi-polymerization and the large size of Cas proteins in the class 1 system, Cas proteins in the class 2 system, especially CRISPR-Cas13 of type VI CRISPR systems, provide promising applications for gene editing and RNA editing.

### 2.2. Properties of CRISPR-Cas13 System in Type VI CRISPR Systems

Recently, researchers identified a CRISPR-Cas13 system in type VI CRISPR systems that can target RNA with high specificity when screening Cas proteins in microbial genome and metagenomics data. To date, the CRISPR-Cas13 system is the only known RNA editing in prokaryotes [12,13,34,35,36]. The Cas13 protein effectors belong to type VI CRISPR systems in the class 2 CRISPR system, and the lack of a RuvC domain of Cas13 effectors is the main difference from other class 2 CRISPR-Cas effectors (type II and V). However, the Cas13 proteins family has two HEPN domains, which can process precursor RNA into mature crRNA and cleave target RNA mediated by crRNA (except for Cas13X).

CRISPR-Cas13 members can be divided into four subtypes (A, B1, B2, C, D, X, and Y) according to their phylogeny. A homology analysis indicated that those four subtypes shared low homology except for the HEPN domain (Figure 2). Interestingly, the most well-studied Cas protein of type VI CRISPR systems is Cas13a, previously called the C2c2 effector [12]. The Cas13a effector (~1000 aa) is the earliest discovered Cas endonuclease that cleaves target ssRNA. Cas13a is a bilobed effector protein, consisting of a crRNA recognition (REC) lobe and a nuclease (NUC) lobe. The REC lobe contains an N-terminal domain (NTD) and a Helical-1 domain, which can bind and recognize crRNA, respectively. The NUC lobe contains two HEPN domains, of which Helical-2 and Helical-3 are separated between two HEPN domains [12]. In 2016, Cas13b (C2c6) was identified via a computational pipeline that searches for CRISPR-Cas locus conserved sequences from the microbial genome and metagenomic data. The size of the Cas13b protein is about 1100 to 1200 aa, and its structure contains a crRNA recognition lobe (REC) and two NUC lobes. Only two HEPN domains of N-terminal and C-terminal from different species are conserved, while other domains show no obvious similarity in amino acid sequence, implying that there are different structures among Cas13b family members [32]. As a member of type VI CRISPR systems, Cas13c (C2c7) likely possesses RNA cleavage activity, but further studies are needed due to there not being enough structural and functional data available.

Yan et al. developed a computational pipeline to construct an expanded database of class 2 CRISPR-Cas systems from microbial genomic and metagenomic data. Then, a new CRISPR-associated ribonuclease was discovered, named Cas13d as a type VI-D CRISPR system [36]. The type VI-D CRISPR-Cas locus was originally identified from gram-positive intestinal bacteria *Eubacterium* and *Ruminococcus*. Cryo-electron microscopy analysis of Cas13d indicated that it was a ~930 aa endoribonuclease and shared less similarity except for two conserved R-X_4–6_ H HEPN motifs (HEPN1 and HEPN2), compared to previous type VI CRISPR systems [14]. Cas13d protein from different species generally owns a bilobed architecture of class 2 effectors. This protein contains five distinct functional domains around the central guide crRNA, which has a REC lobe and a NUC lobe. In the tertiary crystal structure of the complex, the REC lobe possesses an N-terminal domain (NTD) and a Helical 1 domain, while the NUC lobe includes HEPN-1, Helical-1, Helical-2, and HEPN-2 domains. The HEPN-1 domain functions as a structural scaffold to link the REC and NUC lobes [33]. In the main sequence, the structure of Cas13d almost has the same functional domain as Cas13a except for the Helical-1 domain. Besides, Cas13d has a compact structure, and five functional domains of Cas13d are essential for RNase activity, compared to other Cas13 effectors. Moreover, the Cas13d locus as a type VI-D CRISPR system is significantly different from that of other Cas13 effectors (type VI-A and type VI-B) [33]. The crRNA repeat region of Cas13d was relatively conserved in the spacer and its secondary structure. The total length of crRNA is 36 nt with an 8–10 nt stem, 4–6 nt A/U rich ring, a 5–10 nt 5’ terminal, and an AAAAC motif 3’ terminal [14]. Besides, most type VI-D orthologs contain a WYL domain, which enhances both the targeted and the collateral ssRNA cleavage activity in a dose-dependent manner [33,36].

In 2021, Xu et al. developed a computational pipeline to search for previously uncharacterized CRISPR-Cas13 systems from metagenomic datasets. Using CRISPR arrays as search anchors, they identified 340425 putative CRISPR repeat arrays and sequentially analyzed seven Cas13 variants based on conserved stem-loop structure and BLAST alignment similarity, and finally divided them into two groups, including Cas13X (Cas13X.1 and Cas13X.2) and Cas13Y (Cas13Y.1 to Cas13Y.5) [13,37]. The newly identified Cas13X shares some similarity with the previously identified Cas13 family effectors, but possesses a shorter and more compact structure [31]. The crystal structure analysis revealed that the Cas13X protein adopts a bilobed structure consisting of REC and NUC lobes, and the crRNA direct repeat could be anchored in REC leaves. The NUC lobe contains HEPN1 and HEPN2 domains, while the REC lobe is composed of Helical 1, Lid, and Helical 2 domains interleaved with each other. HEPN1 and HEPN2 regions are connected to the REC lobe by inter-domain linkers 1 (IDL1) and 2 (IDL2) [31]. However, currently, no detail structure information of Cas13Y is reported. Cas13bt has been identified as the most ultrasmall family of small Cas nucleases a [38]. The phylogenetic analysis suggested that Cas13bt proteins evolved from larger ancestral Cas13b proteins through multiple deletions. Its crRNA consists of the 5-nucleotide spacer (guide) segment and the 36-nucleotide direct repeat region. The direct repeat region includes stem 1, an internal loop, stem 2, and a hairpin loop. The electron density was less distinct for the spacer region, suggesting its flexibility in the Cas13bt3-crRNA binary complex structure [31].

### 2.3. The Process of Adaptation of the CRISPR/Cas13 Family

The adaptive immune mechanism of type VI CRISPR bacteria is significantly different from that of other types of bacteria due to the differences in the types of foreign nucleic acids. It has been found that, similar to some type III CRISPR systems, Cas1 naturally fuses with reverse transcriptase to form a reverse transcriptase (RT)-Cas1 fusion complex in some type VI CRISPR systems [39,40]. RT associated with the RNA-targeting CRISPR-Cas13 system forms an integrase complex together with Cas1 and Cas2, facilitating the acquisition of RNA molecules, which are integrated into the CRISPR array as a new spacer (Figure 3) [41]. Cas1 proteins play a catalytic role in spacer acquisition from DNA [40].

Recently, it was found that the type VI-A system may include two types of RT-Cas. Type VI-A/RT1 and RT2 systems encompass either one or two CRISPR arrays, each containing two to six spacers of 30–49 nt in length. In type VI-A/RT1 systems, the direct repeats (37-nt) are 6/8-nt longer than those of type VI-A/RT2 systems (29/31-nt). The Cas13a protein sequences of the type VI-A/RT1 and VI-A/RT2 systems clustered separately into two distinct clades with other Cas13a sequences lacking RT-Cas1. Thus, the CRISPR arrays of type VI-A/RT1 and RT2 systems are characteristic of type VI-A systems. Therefore, their association with Cas13a preceded the acquisition of the RT-Cas1/Cas2 adaptation modules [39].

Notably, the RNA immune mechanism of other effectors of the Cas13 family including Cas13b, Cas13c, Cas13X, and Cas13Y are still unclear and need to be studied further.

### 2.4. Function and Molecular Mechanism of CRISPR-Cas13 System

Different subtypes of Cas13 proteins share a common mechanism when binding pre-crRNA and recognizing target RNA molecules (except for the type VI-X and VI-Y). The Cas13 protein recognizes and binds pre-crRNA to form an intermediate transition state.

Functionally, most of the Cas13 effectors are crRNA-guided RNases with two distinct and independent catalytic centers, one of which directly processes pre-crRNAs and the other which cleaves ssRNA. Nakagawa et al. found that pre-crRNA treatment experiments showed that Cas13X did not process its pre-crRNA in vitro [31]. The second catalytic center that can cleave ssRNA possesses two R-X_4_-H motifs, which are typical nucleotide-binding domains in higher eukaryotes and prokaryotes. Generally, the conversion of pre-crRNA to crRNA is processed independently by Cas13 in a metal-independent manner (except for the types VI-D, VI-X, and VI-Y). In Cas13a, the conserved sites of the HEPN-2 domain and Helical-1 domain undergo conformational changes, which enable Cas13a to process pre-crRNA with RNase activity and form a mature Cas13a-crRNA complex. After crRNA maturation, the 5’ end of crRNAs of subtypes VI-A, VI-C, and VI-D contain palindromic repetition regions; however, the crRNA of the VI-B, VI-X, and VI-Y subtypes own the palindromic repetition region at the 3’ end of crRNA. In the second step, in Cas13a, the spacer sequences of target ssRNA and crRNA complement each other and result in a synergistic change to form Cas13a and crRNA complex, which makes the HEPN1 domain closer to the HEPN2 domain. These conformational changes extend the positively charged channel of the NUC lobe to combine crRNA, then unfold the R-X_4_-H motif of the HEPN1 domain for spatial closer access to the second R-X_4_-H motif, resulting in activating the HEPN nuclease site to cleave ssRNA in a nonspecific manner.

Abudayyeh et al. found that the first nucleotide in the protospacer of 3’ flanks preferred A, U, or C (H) rather than G for LshCas13a, which was confirmed by in vitro RNA degradation experiments. G significantly reduced the activity of HEPN nuclease, but A, U, and C enhanced this activity [12,42]. Cas13b-mediated ssRNA cleaves are restricted by a double-side protospacer flanking sequence (PFS), such as a 5′ PFS of D (A, U, or G) and 3′ PFS of NAN or NNA [43].

However, the cleavage of the Cas13d targeted RNA appears to be independent of the original PFS [14]. Cas13d may bind pre-crRNA and cleave it into a 30 nt 5′ palindromic repeat region and a 20 nt 3′ spacer region [36]. Cas13d then recognizes the 5’ direct repeat region of crRNA with the aid of NTD and HEPN2 domains and clamps 2 nt in the head region of the direct repeat region, while the 3’ spacer region is trapped between the Helical-1 and Helical-2 domains. The HEPN1 domain acts as a hinge and provides a structural scaffold to connect the two functional lobes of Cas13d, which may form a positively charged solvent to unwind the RNA spacer. In this compact structure, Cas13d becomes a “surveillance complex” to search and identify complementary RNA target sites [33]. The combination of Cas13d and crRNA mainly occur at the 3’ end of the crRNA repeat region, which is crucial for the correct localization and binding of crRNA and Cas13d [14,33]. Furthermore, the sequence and structure of the crRNA repeat region play an important role in the cleavage activity of Cas13d, and the presence of Mg^2+^ and other auxiliary components, such as the WYL domain, increase its nuclease activity. Recent studies indicated that Mg^2+^ is a key factor that enhances the affinity between Cas13d and crRNA, which may stabilize the conformation of the Cas13d-crRNA repeat region. However, Mg^2+^ does not affect the mature processing of Cas13d on pre-crRNA [33]. EsCas13d and RspCas13d activate RNA cleavage by the WYL1 protein structural domain, indicating a common regulatory mechanism of Cas13d orthologs [36]. Cas13d has a relatively flexible structure, and its WYL1 domain can further stabilize Cas13d by connecting it with crRNA, while the binary Cas13d and crRNA complex recognizes ssRNA and activates Cas13d’s RNA cleavage activity [44]. Due to the activation by the target RNA, all domains of Cas13d further form new conformations and fully integrate with the sugar-phosphate backbone of the crRNA spacer region to form a Cas13d-crRNA-targeted ssRNA ternary complex [45].

The cleavage of the Cas13X targeted RNA is also independent of the original PFS [13]. First, the crRNA DR is recognized by the Helical-1, Lid, and Helical-2 domains. The crRNA is kinked at the DR-spacer junction with the spacer region surrounded by the HEPN1, Helical-1, Lid, and Helical-2 domains. Three nucleotides in the spacer interact with the HEPN1, Lid, and Helical-1 domains within the protein molecule. This structure implied that the DR-proximal region in the spacer cannot serve as a seed sequence that initiates base pairing with a target RNA [31]. Then, in the ternary complex, the crRNA DR is anchored within the REC lobe in the binary complex. The crRNA spacer base matches the target RNA, while the terminal five base pairs are disordered. The crRNA-target RNA duplex is bound to the groove formed by the Helical-1, Lid, HEPN1, and HEPN2 domains [31]. The crRNA-target RNA duplex is recognized by the Helical-1, Lid, HEPN1, and HEPN2 domains through interactions with its sugar-phosphate backbone [31].

## 3. Properties Difference of Cas13 in Type VI CRISPR Systems

Although different subtypes of Cas13 effectors have similar structure and function domains and share common molecular mechanisms, subtle changes in their structures can result in differences in binding pre-crRNA and recognizing target RNA. The structural and functional differences of different Cas13 effectors are listed as follows (Table 2). (1) Due to the difference in primary structure, the protein sizes of Cas13 effectors are significantly different. Among them, Cas13a is about 1250 aa, Cas13b is about 1150 aa, Cas13d is about 930 aa, Cas13Y is about 790 aa, and Cas13X is the smallest, with only about 775 aa. (2) During the process of pre-crRNA, different domains of Cas13a, Cas13b, Cas13d, Cas13X, and Cas13Y participate in crRNA maturation. In Cas13a, the Helical-1 is responsible for the process of pre-crRNA; in Cas13b, the RRI-2 domain is critical for the pre-crRNA process and produces a 66 nt mature crRNA with a 30 nt 5’ spacer and a 36 nt 3’ direct repeat; the compact Cas13d lacks those domains, and the HEPN-2 domain takes on that role. However, Cas13X is not involved in pre-crRNA processing, which is processed by an unknown enzyme [31]. (3) There are different PFS requirements for ssRNA cleavage. For Cas13a, the first nucleotide in the 5’ flanking of ssRNA showed a preference for A, U, or C instead of G. For Cas13b, the selection of ssRNA is diverse based on Cas13b effectors derived from different species. Cas13b effectors from *Bergeyella Zoohelcum* and *Prevotella buccae* prefer 5′ D (A, U, or G) and 3′ NAN or NNA of ssRNA; To date, no PFS sequence has been detected in any Cas13d and Cas13X orthologs, and Cas13Y has not been studied yet. (4) The crRNA structures are different for Cas13a, Cas13b, Cas13d, and Cas13X. For Cas13a, Cas13b, and Cas13d, their crRNA contains a 35–39, 36, or 36 nt direct repeat sequence and forms a conserved 5–6, 3–6, or 8–10 nt stem and a 7–9, 9–14, or 4–6 nt loop, respectively. For Cas13a, Cas13b, and Cas13d, their crRNA contains a 35–39, 36, or 36 nt direct repeat sequence and forms a conserved 5–6, 3–6, or 8–10 nt stem and 7–9, 9–14, or 4–6 nt loop, respectively. However, Cas13X differs slightly from them because the crRNA of Cas13X contains a direct 36 nt repeat sequence and forms stem structures consisting of two separate stems. Stem 1 includes four canonical Watson-Crick base pairs (G-C–G-C), and stem 2 contains a non-canonical G-U wobble base pair and five canonical base pairs (C-G–C-G and C-G), with U flipped out from the stem. The crRNA of Cas13X contains a 5 nt loop [31].

## 4. Application and Prospect of Cas13 in Type VI CRISPR Systems

The VI CRISPR-Cas system has attracted extensive attention due to its advantages of high efficiency, high specificity, programmable RNA targeting, and autonomous pre-crRNA processing. Based on the characteristics of the Cas13 family, researchers developed websites for high-throughput crRNA design: https://gggenome.dbcls.jp/ (22 September 2022) and https://cas13design.nygenome.org/ (22 September 2022). Meanwhile, the RNA editing of Cas13 effectors and their trans-ssRNA cleavage activity accelerate their promising application, such as the clinic diagnosis of pathogens and the treatment of the disease (Table 3). Notably, the plasmid information of the Cas13 system used in the previous studies was summarized in Table 4.

### 4.1. Basic Biochemical Research

Recently, with an in-depth understanding of the structure and function of the CRISPR-Cas13 family, the CRISPR-Cas13 system has been widely used in biological basic research fields, such as gene function, RNA interference, basic medicine, genetic development, etc. Mendez-mancilla et al. found that the chemical modification of crRNA in CRISPR-Cas13d can increase RNA knockdown efficiency in human cells, and demonstrated that the complex of chemically modified crRNA and Cas13d effectors can edit transcript RNA in human primary T cells [49]. Mahas et al. confirmed that CasRx can target a single plant virus or two plant RNA viruses simultaneously with high interference efficiency in *Nicotiana Benthamiana* [50]. Due to RNA-targeting characteristics of the CRISPR-Cas13 system may avoid permanent damage to organisms’ genome, Cas13 can be used as a reliable RNA interference tool for RNA manipulation, including the establishment of RNA editing platform, cell apoptosis and cancer therapy studies, and the manipulation of gene knockdown in animal disease models [45,94,95]. Kushawah et al. developed an efficient, specific, economical, and direct application platform based on CRISPR-RfxCas13d for gene function study during embryogenesis in an animal model with a 76% knockout efficiency [91]. Similarly, Buchman et al. knocked down endogenous gene expression and confirmed its feasibility using a RfxCas13d (CasRx) system in *Drosophila melanogaster* [96].

### 4.2. Nucleic Acid Detection and Diagnosis

Nucleic acid detection and diagnosis technology is a measurement method integrating molecular biology and application techniques. The sensitivity, specificity, quick detection, and low cost are major considered issues for point-of-care testing (POCT). The adaptive immune system of CRISPR-Cas13 members in microorganisms can cleave RNA or trans-cleavage nucleic acids, which may be used for nucleic acid detection and diagnosis (Figure 4a) [47].

Gootenberg et al. designed a fast, cheap, and sensitive portable nucleic acid diagnosis platform, called SHERLOCK, based on the RNA trans-cleavage activity of Cas13 type VI CRISPR systems and strict crRNA complementary base pairing, and successfully applied it for nucleic acid detection and diagnosis [97]. Later, it was gradually applied to the studies of COVID-19, cancer mutations, and plant genetic traits improvement [48,51,52,53]. Currently, researchers are conducting a large-scale clinical study to evaluate SHERLOCK’s availability as a cheap and reliable diagnostic tool for COVID-19 [98,99]. Moreover, other nucleic acid diagnostic tools based on CRISPR-Cas13 have also been developed, including quantitative detection of microRNA and virus surveillance via microfluidic chip. Of note, combinatorial arrayed reactions for the multiplexed evaluation of nucleic acids (CARMEN) can simultaneously perform the high-throughput detection of different species, which may monitor the spread and evolution of infectious diseases [100]. Zhang et al. also developed a hepatitis virus detection technique based on RCA amplification combined with Cas13a, and successfully tested liver tissue samples of hepatitis-infected patients for virus infection with a minimum detection limit of 1 copy/microliter [55]. Due to the COVID-19 pandemic, our society urgently needs fast, labor-saving, and highly efficient diagnostic tools; SHERLOCK has been further simplified as a “streamlined highlighting of infections to navigate epidemics” (SHINE) diagnostic tool for detecting SARS-CoV-2 RNA from unextracted samples [54]. Furthermore, Cunningham et al. modified SHERLOCK and successfully detected plasmodium parasites from different clinical samples in the Democratic Republic of the Congo, Uganda, and Thailand, and achieved 73% sensitivity and 100% specificity [101].

### 4.3. Nucleic Acid Imaging Technology

The visualization of RNA localization and dynamics in living cells is an important technique for the in-depth study of RNA subcellular localization and function. Currently, single-molecule fluorescence in situ hybridization (smFISH) of RNA is the most commonly used method to observe the dynamics of intracellular RNA. However, CRISPR-Cas13 tagging fluorescent markers accelerate the development of RNA tracking and imaging in living cells (Figure 4b) [56,102]. In 2017, for the first time, the Cas13 family effector Cas13a was applied for intracellular nucleic acid imaging, in which dLwaCas13a-NF can image stress granule formation in living cells and realize the dynamic observation of intracellular transcripts [46]. The dPspCas13b and dPguCas13b fusing different fluorescent proteins successfully achieved quick and efficient tracking of target RNA [58,89]. Besides, dRfxCas13d combined with fluorescence-labeled crRNA may observe RNA transcription in living cells using CRISPR Live-cell fluorescent in situ hybridization (LiveFISH) [56]. Moreover, a new system based on dCas13a-SunTag-BiFC was developed by fusing dLwaCas13a and SunTag systems. In this system, dLwacas13a was used as a tracker targeting specific RNA, while SunTag recruits split Venus fluorescent proteins to label the targeted RNA [60].

### 4.4. Antiviral Applications

Immunotherapy and precision medicine based on RNA editing become effective alternatives for antiviral therapies in animals and plants due to their specific target delivery and non-permanent genetic effect [103]. Previous studies indicated that ssRNA viruses infecting humans, vertebrates, and plants occupied approximately 51%, 44%, and 70% of all viruses, respectively, implying that the application of Cas13 targeting ssRNA is promising and has a broad market potential (Figure 4c) [50,103].

For antiviral therapy in mammals and plants, Freije et al. showed that the Cas13 effector can be programmed to target and destroy the genomes of multiple mammalian single-stranded RNA viruses, including lymphocytic choriomeningitis virus (LCMV), influenza A virus (IAV), and vesicular stomatitis virus (VSV) [104]. Moreover, the strategy of Cas13-mediated RNA virus interference is expected to be an effective tool for plant immunity against plant RNA viruses. Using the CRISPR/Cas13a system, Zhang et al. constructed an antiviral protection tool for monocotyledons, which can target and cleave viral RNA genomes to resist RNA viruses [63]. Abbott et al. developed prophylactic antiviral CRISPR in human cells (PAC-MAN) for simultaneously targeting multiple coronaviruses and influenza viruses via highly conserved regions of the virus genome [61]. Cui et al. developed an all-in-one expression system expressing Cas13b and double crRNA to meet the needs for synchronal delivery of Cas13b proteins and crRNA, which can eliminate PRRSV infection [62]. Because of the rapid spread of novel coronavirus, some researchers have proposed RNA virus treatment strategies based on CRISPR/Cas13 family effectors for coronavirus infection [82,105]. Ashraf et al. have preliminarily achieved the treatment of hepatitis C in mammalian cells [64]. Zhang et al. developed a powerful tool based on the CRISPR/Cas13d system to treat Seneca Valley virus (SVV). In this system, CasRx-de-NSL can be expressed in human cells, and SVV RNA can be cleaved and treated. The results showed that the experimental group reduced the replication ability of SVV by 57% [106]. As the smallest type VI nucleic acid effector, Cas13X is an excellent candidate for treating genotypic diseases. Yang et al. utilized the AAV-PHP.eB capsid and GFAP promoter to drive the specific expression of Cas13X-NLS-HA-sgPtbp1 in astrocytes and showed that the PTBP1 signal, a key signal, was decreased after two weeks, one month, and two months of intravenous treatment, respectively. The PTBP1 S signal decreases sequentially from 74.79% to 18.42%, 14.96%, and 11.98%, indicating that astrocytes can be successfully converted into neurons (AtN) [107].

### 4.5. Disease Treatment Strategies Based on CRISPR/Cas13

CRISPR/Cas13-based gene therapy strategies can treat many diseases, such as dominant and recessive genetic diseases, cancer, cardiovascular diseases, and neurodegenerative diseases [108]. The CRISPR-Cas13 system may knock down intracellular RNA to block the occurrence of mutagenic and genetic diseases. Cas13d, the smallest Cas13 protein, has developed a clinic platform for RNA therapy (Figure 4d). To better apply the CRISPR/Cas13 system for the treatment of human and animal diseases, researchers have developed multiple plasmids and RNA delivery techniques, including the intravenous injection of Cas protein and crRNA, the stable delivery of AAV lentiviral vector, and PB transposon systems fused with CRISPR/Cas13 and crRNA [69]. Zhao et al. found that the subtly designed crRNA combined with Cas13a could inhibit the transcriptional expression level of the KRAS mutant in pancreatic cancer, and ultimately suppressed oncogene expression in mice, suggesting that the CRISPR/Cas13 system may be useful for tumor therapy by perturbing oncogenes at the transcriptional level [74]. He et al. proposed that the delivery of AAV (adenovirus) mediated CasRx and Pcsk9 sgRNA into the mice liver could successfully reduce serum Pcsk9 and cholesterol levels, which provided a permanent genetic therapy using Cas13 proteins [109]. Due to the continuous outbreak of SARS-CoV-2, researchers have also applied CRISPR/Cas13 for COVID-19 treatment [109,110,111]. Xiao et al. evaluated the therapeutic potential of a Cas13-derived RNA base editor to correct mutations in myosin VI (*Myo6*) transcripts that cause hearing loss in a mouse model [66].

In previous studies, oncogenes and toxic RNAs were knocked down by Cas13a under the control of a minimal promoter regulated by NF-κB transcription factor binding [112]. In addition, in the gene therapy study of Cas13a, the researchers induced EGFR overexpression by regulating the key oncogenic genes in glioma cells and caused the apoptosis of glioma cells [113]. Tian et al. used an alternative polyadenylation (APA) reporter to screen a set of dCas13 proteins, including Cas13a, Cas13b, and Cas13d, and showed that the CRISPR- dPguCas13b system had the most significant efficiency of APA manipulation [114].

### 4.6. Other Applications

Cas13 family proteins are also widely used for the mRNA process because they are currently found to be CRISPR effectors that specifically cleave RNA [115]. The first other application is to participate in alternative splicing (SA) and APA of mRNA maturation by the CRISPR-Cas13 system. Konermann et al. reported that CRISPR-dCas13d targeted splicing elements and engineered fusion into the Gly-rich C-terminal domain of hnRNPa1, one of the most abundant hnRNP families, which can successfully interfere with exon exclusion or inclusion in endogenous gene reporter systems. This demonstrated that the CRISPR-Cas13 system is a useful tool for the gene functional study of AS events [14]. Second, the CRISPR-Cas13 system can be applied for RNA modification, including programmable regulation of alternative splicing, A-to-I and C-to-U editing, and m6A modifications [89,116,117]. Third, the CRISPR-Cas13 system can be used for the design of cell fate. Previous studies utilized the single-base editing technology of CRISPR/dCas13 to convert C to U for nucleic acid mutations. Furthermore, STAT3 and β-catenin pathways were activated via the CRISPR-Cas13 system to accelerate the growth of HEK293FT cells and HUVECs [118]. In addition, it was reported that the demethylation of m^6^A in *PTH1R* mRNA by the photoactivatable RNA m^6^A editing system using CRISPR-dCas13 (PAMEC) in bone marrow mesenchymal stem cells could hinder osteogenic differentiation and reduce translation efficiency without changing RNA stability [119].

## 5. Future Perspectives

As we know, because RNA is one of the most important messengers in cellular processes, characterization of its function and dynamic metabolism is crucial for understanding life processes. The use of the RNA-targeted CRISPR-Cas13 system will contribute to the understanding of the RNA’s function. Cas13 protein is an effector with dual ribonuclease activities (including pre-crRNA processing and crRNA-mediated RNA cleavage), and has unique advantages in RNA-specific recognition, RNA cleavage, and trans-activating activity). It has a wide application prospect in biomedical basic research, animal disease models, genetic disease diagnosis, nucleic acid subcellular localization, and disease treatment. Presently, although the CRISPR-Cas13 family can avoid permanent damage to the genome of organisms because of RNA targeting, it should be considered for the toxic effects of RNA-targeted cleavage. Cas13X in type VI CRISPR systems possesses a small size (~775 aa) without a PFS limit, enabling it to be conveniently fused with multiple RNA arrays and improving its high-throughput application. Taken together, RNA editing targeted by the CRISPR-Cas13 family further expands the CRISPR toolkit for RNA manipulation and greatly enhances diagnostic capabilities.

## Figures and Tables

**Figure 1 ijms-23-11400-f001:**
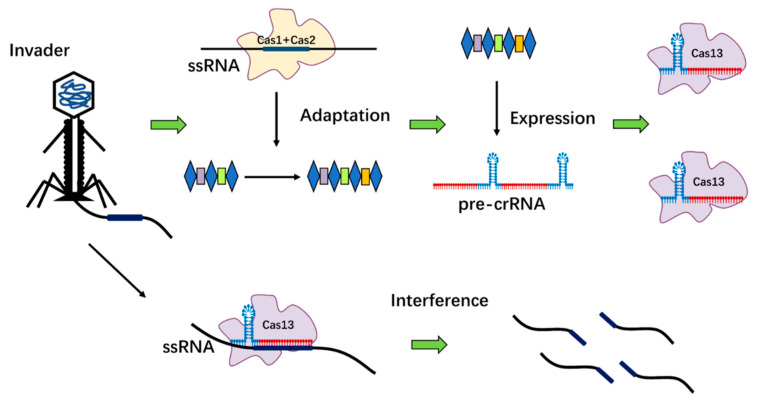
Mechanism of immunity against virus invaders through CRISPR-Cas systems. The molecular defense process of the CRISPR-Cas system includes three phases: adaptation, maturation, and interference. Adaptation: the host organism captures the nucleotide fragments from invaders and integrates them into the CRISPR array; Maturation: CRISPR array is transcribed into precursor crRNA (pre-crRNA) and further processed into mature crRNA; Interference: crRNA binds to the effector Cas protein to form a complex, and the target invader genome (DNA/RNA) is identified and bound with the help of crRNA, which eventually leads to the degradation of the invader genome (DNA/RNA).

**Figure 2 ijms-23-11400-f002:**
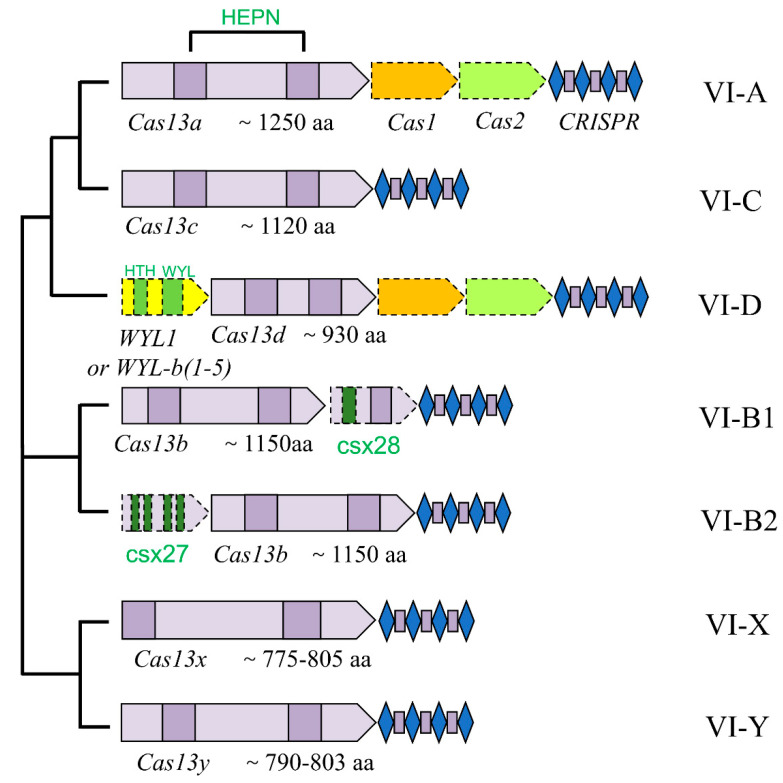
The architecture of CRISPR locus of VI CRISPR-Cas systems and their phylogenetic relationships. The class 2 systems (type VI CRISPR systems) consist of a single, comparatively larger Cas effector protein and CRISPR locus, which can be divided into four subtypes VI-A, VI-B1, VI-B2, VI-C, VI-D, VI-X, and VI-Y. Different colors represent different functional domains in this phylogenetic tree. The HEPN domains of each effector are represented by dark purple squares separated by other structural units. Grey rectangles denote CRISPR direct repeats (DRs), and dark purple squares indicate spacer sequences. The size of each effector and its corresponding conserved domain are indicated as the gene box.

**Figure 3 ijms-23-11400-f003:**
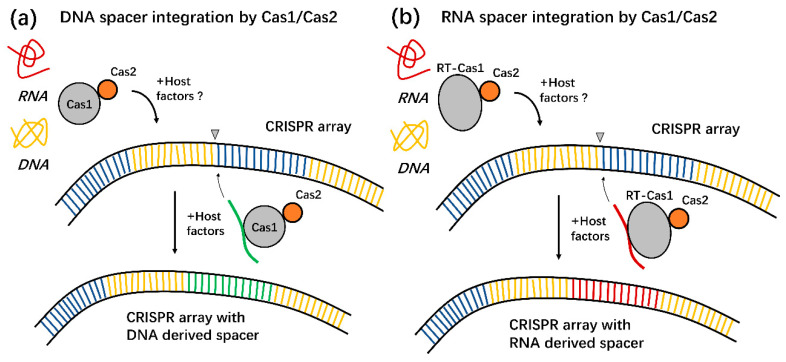
Schematic diagram of DNA and RNA spacers acquisition by CRISPR-Cas systems. (**a**) Small segments of invasive DNA are assimilated into CRISPR arrays by Cas1 and Cas2 in a canonical spacer acquisition process that allows adaptive immunity in a wide variety of bacteria and archaea. (**b**) In the type Ⅵ CRISPR systems, an RT fused to Cas1 enables the acquisition of spacer sequences directly from RNA. This process can mediate adaptive immunity against RNA-based parasites [40].

**Figure 4 ijms-23-11400-f004:**
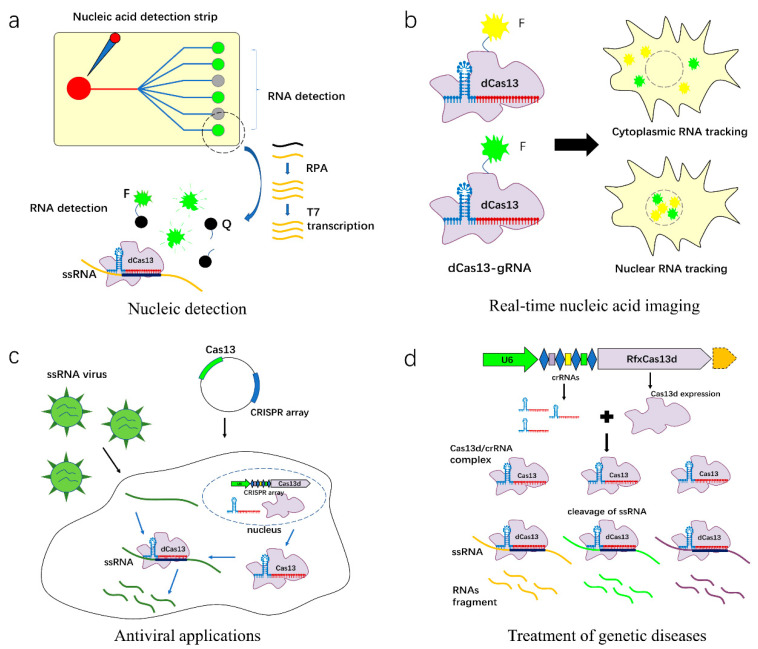
Applications of CRISPR-Cas13 system. (**a**). Cas13-driven biosensor for disease diagnosis. Cas13-based DNA and RNA detection methods and sample preparation steps are integrated into microfluidic biosensors for rapid pathogen detection; (**b**). Real-time nucleic acid imaging and tracking of living cells using type VI CRISPR systems. dCas13 proteins fused with fluorescent proteins (FPs) effector and coupling with specific guide RNA can be used for spatiotemporal visualization of the target RNA transcripts in living cells; (**c**). Antiviral application of type VI CRISPR systems. The type VI CRISPR systems target the genomic region of evolutionarily conserved ssRNA viruses to inhibit viral infection in human animals and plants; (**d**). CRISPR-Cas13 system-mediated genetic disease treatment.

**Table 1 ijms-23-11400-t001:** Comparison of the Cas9 and Cas13 systems.

Cas Effectors	Cas9	Cas13
Protein size	~1100 aa	~775–1250 aa
Targeted substrate	dsDNA	RNA
PAM/PFS	The PAM region of spCas9 was NGG, while that of saCas9 was NNGRRT	Different Cas13 family types have different PFS requirements, among which Cas13a tends to A, U, and C, Cas13b tends to A, U, and G, while Cas13d and Cas13X have not been found to have PFS.
Targeted site distribution	Due to the limitation of the PAM region of Cas9, the distribution of editable sites is relatively general.	Due to the limitation of the PAM region of Cas13a and Cas13b, the distribution of editable sites is relatively general. However, for Cas13d and Cas13X, no obvious PFS region has been found, which has an extensive selection of editing sites.
Trans-cleavage	No trans-cleavage activity was found	Cas13a, Cas13b, and Cas13X were found to have significant trans-cleavage activity.
Damage to organisms	There’s permanent damage to the genome	Because the Cas13 family effectors target editing at the RNA level, they rarely cause permanent genetic damage to cells or organisms.
Application	Knockin and knockout of genes; single base mutation in DNA; epigenetic regulation	Basic biochemical research; nucleic acid detection and diagnosis; nucleic acid imaging technology; antiviral applications; disease treatment strategies based on CRISPR/Cas13

**Table 2 ijms-23-11400-t002:** Characteristics of Cas13 effectors in type VI CRISPR systems.

Cas Effectors	Cas13a (VI-A)	Cas13b (VI-B)	Cas13d (VI-D)	Cas13x (VI-X)
Cas protein size	~1250 aa	~1150 aa	~930 aa	~775–805 aa
Protospacer-flanking site (PFS)	A, U and C	A, U and G	no PFS constraints	no PFS constraints
Architecture	REC and NUC lobes	pyramidal (binary complex)	REC and NUC lobes	REC and NUC lobes
pre-crRNA processing site	Helical-1 and HEPN-2 domains	RRI-2 domain	HEPN-2 domain	not process
Direct repeat lengths	35–39 nt	36 nt	36 nt	36 nt
Orientation (repeat to spacer)	5′–3′	3′–5′	5′–3′	3′–5′

**Table 3 ijms-23-11400-t003:** Cas13-based RNA technologies.

Application Field	Cas Effectors	Efficiency	Application
Basic biochemical research	Cas13a	Medium	RNA knockdown, Nucleic acid detection [46]
	Cas13a	2 × 10^3^ copies/mL	Nucleic acid detection (SHERLOCK) [47]
	Cas13a	High (90%)	Virus interference, transcript targeting guide-induced, gene silencing [48]
	Cas13d	High (89%)	Chemically modified crRNAs can modify the transcriptome of human primary T cells [49]
	Cas13a	40.4–83.9%	Demonstrates the RNA-guided RNase activity of the Cas13a [46]
	Cas13b		Discovery and biochemical activity of Cas13b [43]
	Cas13a, Cas13b		RNA interference, virus interference, and virus resistance [50]
	Cas13d	34–46%	Therapeutic potential, generation of AAV all-in-one vector consisting of up to three pre-sgRNAs for effective knockdown of VEGFA gene expression [45]
	Cas13d		Discovery and biochemical activity of Cas13d [36]
	Cas13X		Discovered Cas13x effector; exhibited robust editing efficiency and high specificity to induce RNA base conversions [13]
Nucleic acid detection	Cas13a	Medium	Nucleic acid detection (SHERLOCKv2) [47]
	Cas13a	High	One-step experimental screening system, diagnostics, and therapeutics for COVID-19 [51]
	Cas13a	High (100%)	RT-LAMP, point-of-care diagnostics, detect SARS-CoV-2 [52]
	Cas13a		CRISPR diagnostics and targeted cancer therapy [53]
	Cas13a	High (90%)	Lateral flow strip, single-step SARS-CoV-2, two-step SARS-CoV-2 assay (SHINE) [54]
	Cas13a		Hepatitis B virus covalently closed circular DNA detection [55]
Nucleic acid imaging	Cas13d		CRISPR Live-cell fluorescent in situ hybridization (LiveFISH) accurately detects chromosomal disorders and tracks the real-time movement of DNA double-strand breaks [56,57]
	Cas13b		Dynamic imaging of RNA in living cells, simultaneous visualization of RNA-RNA and DNA-RNA in living cells [58]
	Cas13b		RNA-protein interactions identify proteins associated with an endogenous RNA, CRISPR-based RNA proximity proteomics (CBRPP) [59]
	Cas13a		Endogenous RNA foci imaging of RNA in the nucleus and cytoplasm in living cells [60]
Antiviral application	Cas13d	90%	CRISPR-based strategy for RNA-guided viral RNA inhibition and degradation (PAC-MAN) [61]
	Cas13b	50%	Abrogation of pRRSV infectivity in mammalian cells [62]
	Cas13a		RNA virus resistance in both dicot and monocot plants [63]
	Cas13a	70–84%	CRISPR-Cas13a mediated targeting of hepatitis C virus internal-ribosomal entry site (IRES) [64]
Disease treatment	Cas13d		Using pgRNAs can robustly suppress the propagation of plant RNA viruses [65]
	Cas13a		GIGS offers a novel and flexible approach to RNA reduction for crop improvement and functional genomics [48]
	Cas13X		Rescue of autosomal dominant hearing loss by in vivo delivery [66]
	Cas13b		RNA base editing; an efficient RNA base editor, dPspCas13b-RESCUE-NES, a potentially useful tool for biomedical research and genetic disease [67]
	Cas13a, Cas13b, Cas13d		A one-step platform for screening high-efficient and minimal off-target CRISPR/Cas13 crRNAs to eradicate the SARS-CoV-2 virus for the treatment of COVID-19 patients [51]
	Cas13a		Detects BK polyomavirus DNA and cytomegalovirus DNA from patient-derived blood and urine samples [68]
	Cas13d	100%	PiggyBac systems; Cas13d vector achieved extremely high efficiency in RNA knockdown (98% knockdown for CD90) with optimized gRNA designs [69]
	Cas13d	99%	Alleviation of neurological disease by RNA editing [70]
	Cas13d		Establish RfxCas13d as a versatile platform for knocking down gene expression in the nervous system [71]
	Cas13	50%	A versatile tool for cancer diagnosis, therapy, and research [72]
	Cas13		Cas13s for targeting viral RNA [53]

**Table 4 ijms-23-11400-t004:** Plasmid information of Cas13 system used in the previous studies.

Cas Effectors	Name of Plasmid Vector	Use	References
Cas13a	pET-Sumo-LbuCas13a	Expression plasmid	[73]
	pCas13a-gRNA	Antiviral strategy	[64]
	pET-Lsh.Cas13a vector	Disease treatment	[74]
	pC013-Twinstrep SUMO-huLwCas13a	Detection and diagnosis	[75]
	pMD19T-E	Detection and diagnosis	[76]
	pC016-LwCas13a-GFP	Disease treatment	[77]
	pC016-LwCas13a-Ctrl	Disease treatment	[77]
	pC016-LwCas13a-RdRP	Disease treatment	[77]
	pC016-LwCas13a-PPIB	Disease treatment	[77]
	pC016-LwCas13a-CXCR4	Disease treatment	[77]
	pC016-LwCas13a-KRAS	Disease treatment	[77]
	pC016-LwCas13a-N	Disease treatment	[77]
	pET28a-Cas13a-XLCHN-DTR-His	Disease treatment	[77]
	pK2GW7-pCas13a vector	Plant resistance	[78]
	pC016 LwCas13a	Cancer treatment	[79]
	pC034-LwCas13a-msfGFP-2A-Blast	Cancer treatment	[79]
	Lentiviral vector (unnamed)	Cancer treatment	[80]
	pET-Sumo-LbuCas13a expression vector	miRNA detection	[81]
	pDUAL-HFF1-Cas13a expression vectors	Retrovirus interference	[82]
	pKS-rrk1-(LshCas13a crRNA)-Control	Retrovirus interference	[82]
	pET-Sumo-LbuCas13a expression vectors	Basic research	[83]
	pUb LwaCas13a + LwaCas13a Guide RNA	Expresses LwaCas13a and guide RNA	[84]
	pLsCas13aGG	Backbone plasmid	[85]
	puc19-pCas13a	Intermediate/cloning vector	[86]
	pC015-dLwCas13a-NF	Expresses negative	[46]
	pC014-LwCas13a-msfGFP	Expresses active LwCas13a	[46]
	pC035-dLwCas13a-msfGFP	Expresses catalytically inactive LwCas13a	[46]
	pGJK_His-SUMO-LbuCas13a	Bacterial expression	[87]
	pDuBir-Lbu-dCas13a-avitag	Dual expression of Lbu-dCas13a and BirA	[88]
	pC0056-LwCas13a-msfGFP-NES	Expresses active LwaCas13a-NES	[46]
	pC034-LwCas13a-msfGFP-2A-Blast	Expresses active LwCas13a	[46]
	p2CT-His-MBP-Lwa_Cas13a_WT	Bacterial expression for Cas13a	[42]
	p2CT-His-MBP-Lne_Cas13a_WT	Bacterial expression for Cas13a	[42]
	p2CT-His-MBP-Lba_Cas13a_WT	Bacterial expression for Cas13a	[42]
	p2CT-His-MBP-Ere_Cas13a_WT	Bacterial expression for Cas13a	[42]
	p2CT-His-MBP-Cam_Cas13a_WT	Bacterial expression for Cas13a	[42]
	p2CT-His-MBP-Rca_Cas13a_WT	Bacterial expression for Cas13a	[42]
	p2CT-His-MBP-Hhe_Cas13a_WT	Bacterial expression for Cas13a	[42]
	p2CT-His-MBP-Ppr_Cas13a_WT	Bacterial expression for Cas13a	[42]
	pHAGE-IRES-puro-NLS-dLwaCas13a-EGFP-NLS-3xFlag	Overexpression	[58]
	pHAGE-IRES-puro-NLS-dLbaCas13a-EGFP-NLS-3xFlag	Overexpression	[58]
Cas13b	pUb PspCas13b + PspCas13b guide RNA	Expresses PspCas13b	[84]
	pBzCas13b/pPbcas13b/pBzCas13b/pBzCas13b-HEPN	Bacterial expression for Cas13b	[43]
	pC0041-RanCas13b crRNA backbone	For cloning of guide RNAs compatible with RanCas13b	[89]
	pU6-PspCas13b-gRNA-Actb1216	PspCas13b guide RNA	[90]
	pAB1620 hU6-BpiI-Cas13bt3-DR	hU6-BpiI-Cas13bt3-DR (crRNA expression)	[38]
Cas13d	pUb RxCas13d + RxCas13d guide RNA	Expresses RxCas13d	[84]
	pUb dRxCas13d + RxCas13d guide RNA	Expresses catalytic dead RxCas13d	[84]
	pT3TS-RfxCas13d-HA	Plasmid to carry out IVT of RfxCas13d	[91]
	pET28a-MH6-EsCas13d	Expresses E. coli codon-optimized EsCas13d	[36]
	pET28a-MH6-RspCas13d_RspCasWYL1	Expresses E. coli codon-optimized RspCas13d and RspCasWYL1	[36]
	pT3TS-RfxCas13d-NLS-HA	Plasmid to carry out IVT of RfxCas13d-NLS	[91]
	dCas13d-dsRBD-APEX2	TetON-APEX2-V5-BPNLS-dRfxCas13d-dsRBD-BPNLS-P2A-GFP	[92]
	pET-28b-RfxCas13d-His	Plasmid for bacterial expression and purification of RfxCas13d protein	[91]
	pSLQ5428_pHR_EF1a-mCherry-P2A-Rfx_Cas13d-2xNLS-3xFLAG	Lentiviral vector encoding Rfx Cas13d fused with 2xNLS, 3xFLAG, and 2A-tagged mCherry	[61]
	pLentiRNAGuide_002-hU6-RfxCas13d-DR-BsmBI-EFS-Puro-WPRE	For cloning of guide RNAs libraries compatible with RfxCas13d	[93]
Cas13X	CMV-Cas13X.1-SV40pA_U6-BbsI-DR_CMV-mCherry-BGHpA	Expression vector for encoding a human codon-optimized Cas13X.1 driven by CMV promoter	[13]
	CMV-dCas13X.1-REPAIRv2-SV40pA_CMV-mCherry-BGHpA_U6-BbsI-DR	Expression vector for encoding a human codon-optimized dCas13X.1-REPAIRv2 driven by CMV promoter	[13]
	U6-BbsI-DR_CMV-minidCas13X.1-REPAIRv2-BGHpA_CMV-EGFP-BGHpA	Expression vector for encoding a human codon-optimized minidCas13X.1-REPAIRv2 driven by CMV promoter	[13]
	CMV-dCas13X.1-RESCUE-S-SV40pA_U6-BbsI-DR_CMV-mCherry-P2A-Puro-BGHpA	Expression vector for encoding a human codon-optimized dCas13X.1-RESCUE-S driven by CMV promoter	[13]
	CMV-minidCas13X.1-RESCUE-S-SV40pA_U6-BbsI-DR_CMV-mCherry-P2A-Puro-BGHpA	Expression vector for encoding a human codon-optimized minidCas13X.1-RESCUE-S driven by CMV promoter	[13]

## Data Availability

Not applicable.

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
