# Peer review of "Insights Gained from RNA Editing Targeted by the CRISPR-Cas13 Family"

_ijms, 2022, doi:10.3390/ijms231911400_

Round 1
Reviewer 1 Report
In the present review Li and Sheng, compiled the information on the updates of in RNA editing targeted by the CRISPR-Cas13 family, and discussed the application of Cas13 in basic research, nucleic acid diagnosis, nucleic acid tracking, and genetic disease treatment. However, it comes up with reservations and concerns, my suggestions and comments which you can find below.
First in terms of information, this review should be more comprehensible. The review appears to be primarily focused on a basic class of cas13. Because various studies are missing, the review is quite brief.
There should be at least one table indicating that the RNA editing has been completed and its success rate using the cas13 system.
The study demands the pros and cons for CRISPR Cas9 and 13 comparison in tabular format.
This review manuscript discusses the 5 application of Cas13 system, does it means only these application are possible? I would recommend to add the another heading with “others applications” to report and describe the supplementary application and possibility of field where it can be used.
Authors should include the table for this review article describing what kind of studies have been performed till now using Cas13 system.
It would be nice resource if authors provide the information in vectors list that are available to be used for Cas13 system.
Author Response
Dear reviewer,
Thank you for reviewing our manuscript “Insights gained from RNA editing targeted by the CRISPR-Cas13 family” We appreciate the comments from two anonymous reviewers. According to the comments, the manuscript has been carefully corrected to improve its quality. The point-to- point responses are listed below.
Reviewer: #1: Comments and Suggestions for Authors
In the present review Li and Sheng, compiled the information on the updates of in RNA editing targeted by the CRISPR-Cas13 family, and discussed the application of Cas13 in basic research, nucleic acid diagnosis, nucleic acid tracking, and genetic disease treatment. However, it comes up with reservations and concerns, my suggestions and comments which you can find below.
1.First in terms of information, this review should be more comprehensible. The review appears to be primarily focused on a basic class of cas13. Because various studies are missing, the review is quite brief.
Response: Thank you so much for the constructive comments! We carefully revise this review according your suggestion. Now, this review briefly describes the Cas13 family effectors, including their classification, structure, function, mechanism of action, and applications. Among them, Cas13a, Cas13d, and Cas13X are mainly introduced in detail. We add some important information, including the immune mechanism of Cas13 family proteins (Part II), the structure, function and application of newly discovered Cas13X and Cas13Y, and other related applications.
2.There should be at least one table indicating that the RNA editing has been completed and its success rate using the cas13 system.
Response: In new revision, we add the Table 3 about Cas13 gene editing completed and complement the information about its success rate.
3.The study demands the pros and cons for CRISPR Cas9 and 13 comparison in tabular format.
Response: In the new version, we add the Table 1 about the advantages and disadvantages of Cas9 and Cas13.
4.This review manuscript discusses the 5 application of Cas13 system, does it means only these application are possible? I would recommend to add the another heading with “others applications” to report and describe the supplementary application and possibility of field where it can be used.
Response: We first enriched 5 applications of the Cas13 system, which are mainly included in 4.2. We added two applications of nucleic acid detection and diagnosis, “Zhang et al. also developed a hepatitis virus detection technique based on RCA amplification combined with Cas13a, and successfully tested liver tissue samples of hepatitis infected patients for virus infection with a minimum detection limit of 1 copy/μL (Zhang et al., 2022A)” and "Cunningham et al. modified SHERLOCK and successfully detect Plasmodium from different clinical samples in the Democratic Republic of the Congo, Uganda, and Thailand, achieving 73% sensitivity and 100% specificity (Cunningham et al., 2021)."
In the application section of nucleic acid imaging, we have added some important references, including "In 2017, for the first time, the Cas13 family Effector Cas13a was applied to intracellular nucleic acid imaging, in which dLwaCas13a-NF can image stress granule formation in living cells and realize dynamic observation of intracellular transcripts (Abudayyeh et al., 2017)." and "A new system based on dcas13a-SunTag-BiFC was developed by fusing dLwacas13a and SunTag systems. dLwacas13a is used as a tracker targeting specific RNA, While SunTag recruits the split Venus fluorescent proteins to label the targeted RNA (Chen et al., 2022)".
In the issue of antiviral application, we also added some recent research results, mainly modified as follows: " Because of the rapid spread of novel coronavirus, some researchers have proposed RNA virus treatment strategies based on CRISPR/Cas13 family effectors for coronavirus infection (Nguyen et al., 2020; Zhang, 2020). Ashraf et al. have preliminarily realized the treatment of hepatitis C in mammalian cells (Ashraf et al., 2021). Zhang et al. evaluated a powerful tool based on the CRISPR/Cas13d system to treat the Seneca Valley virus (SVV). In this system, CasRx-de-NSL can be expressed in human cells, and SVV RNA can be cleaved and treated. The results showed that the experimental group reduced the replication ability of SVV by 57% (Zhang et al., 2022b). As the smallest type VI nucleic acid effector discovered so far, Cas13X is an excellent candidate for use in the treatment of genotypic diseases. Yang et al. applied AAV-PHP.eB capsid and GFAP promoter to drive the specific expression of Cas13X-NLS-HA-sgPtbp1 in astrocytes and showed that the PTBP1 signal was decreased after 2 weeks, 1 month, and 2 months of intravenous treatment, respectively. The PTBP1 S signal decreases sequentially from 74.79% to 18.42%, 14.96%, and 11.98%, which indicates that astrocytes can be successfully converted into neurons (AtN) (Yang et al., 2022)".
In terms of the application of disease treatment strategies, we added some recent research results, which are mainly modified as follows: “In some studies, Cas13a knockdown of oncogenes and toxic RNA was achieved by placing it under the control of a minimal promoter regulated by NF-κB transcription factor binding (Gao et al., 2020). Besides, in the gene therapy study of Cas13a, the researchers induced EGFR overexpression by regulating the key oncogenic genes in glioma cells, thereby realizing the apoptosis of glioma cells (Wang et al., 2019b). Tian et al. used Alternative polyadenylation (APA) reporter to screen a set of dCas13 proteins, including Cas13a, Cas13b, and Cas13d, and showed that the CRISPR- dPguCas13b system had the most significant efficiency of APA manipulation. It can also engineer the use of endogenous PAS and is an effective APA micromotion tool for studying disease-related APA events (Tian et al., 2022a)”.
Moreover, besides these 5 applications, this revision adds some other applications, which is shown in “4.6. Other application”.
5.Authors should include the table for this review article describing what kind of studies have been performed till now using Cas13 system.
Response: We added the Table 3 in the revised text, mainly describing the application types of the Cas13 system.
6.It would be nice resource if authors provide the information in vectors list that are available to be used for Cas13 system.
Response: Thank you so much for reviewing our manuscript! In the revised version, we have added Table 4 of the plasmid vector list, which describes the carrier information used in the previous studies on the Cas13 system.
Reviewer 2 Report
The authors try to present current knowledge about molecular biology and applications of CRISPR/Cas13 system. However, the review contains a little of novel information.
Major points.
A number of recent reviews on the CRISPR/Cas13 system appeared in 2021 and 2022, which also cover all aspects of its molecular biology and applications, e.g. DOI: 10.3390/bioengineering9070291, 10.3389/fcell.2021.677587, etc.
In this review, only 3 references out of 65 are from 2022, and 8 references are from 2021. Therefore, this review adds very little to the current knowledge of this system. Thus, the authors need to find and discuss many more references and data presented on this system in 2021 and 2022 to increase the novelty of the presented data.
For example, the authors did not describe and discuss the recently found Cas13X and Cas13Y systems. May be other systems has been found?
Section 2 The authors need to describe the current data on how the CRISPR/Cas13 system initially recognizes target RNAs, converts them to DNA, and inserts the appropriate spacers into the CRISPR loci.
Section 2.2 Authors need to describe the structure of the recently discovered VI-X (Cas13X) and VI-Y (Cas13Y) systems (e.g. DOI: 10.1038/s41592-021-01124-4).
Section 4 Authors need to describe the application of the recently discovered VI-X (Cas13X) and VI-Y (Cas13Y) systems. Cas13d is not the smallest protein of type IV system anymore. The authors did not describe applications of Cas13 effectors in possible therapeutic application by the regulation of mRNA modifications, splicing and translation.
In line 34, remove the duplicate word "identified" in the phrase "first identified and identified a."
Minor points.
In the introduction, the authors need to mention the cornerstone work of Martin Ginek et al. DOI: 10.1126/science.1225829, where they elegantly showed the complete mechanism of action of CRISPR/Cas9. This work has been followed by others showing that CRISPR/Cas9 can be applied to edit the human genome and other organisms.
Line 45 Since the authors are talking about two articles, they need to refer to the second article DOI: 10.1126/science.1232033.
Lines 45-47. The authors need to be more specific: Cpf1(Cas12a) recognizes AT-rich PAMs, so it can target AT-rich regions where can be no GC-rich PAMs for Cas9, but Cpf1(Cas12a) has difficulty finding a target in GC-rich regions. Therefore, Cpf1(Cas12a) together with Cas9 expands the number of possible target regions.
Line 49 The authors should be more precise in their terminology. "Genetic DNA of organisms" is an awkward term. It is better to say DNA.
Author Response
Dear reviewer,
Thank you for reviewing our manuscript “Insights gained from RNA editing targeted by the CRISPR-Cas13 family” We appreciate the comments from two anonymous reviewers. According to the comments, the manuscript has been carefully corrected to improve its quality. The point-to- point responses are listed below.
Reviewer: #2: Comments and Suggestions for Authors
The authors try to present current knowledge about molecular biology and applications of CRISPR/Cas13 system. However, the review contains a little of novel information.
Major points.
- A number of recent reviews on the CRISPR/Cas13 system appeared in 2021 and 2022, which also cover all aspects of its molecular biology and applications, e.g. DOI: 10.3390/bioengineering9070291, 10.3389/fcell.2021.677587, etc.
Response: Thank you so much for valuable suggestions! During the process of writing this paper, we also noticed that some researchers published review articles on the Cas13 system. Thus, our review aims to show some different views and contents from other researchers. Meanwhile, due to the newly discovered Cas13X and Cas13Y systems proposed by the reviewer, we have also added this part into this review, which will become a relatively complete and comprehensive review.
- In this review, only 3 references out of 65 are from 2022, and 8 references are from 2021. Therefore, this review adds very little to the current knowledge of this system. Thus, the authors need to find and discuss many more references and data presented on this system in 2021 and 2022 to increase the novelty of the presented data.
Response: In the revised manuscript, we have re-added some important references published in 2021 and 2022, including 23 references published in 2021 and 17 references published in 2022. The main contents include studies on Cas13X and Cas13Y, the application of the Cas13 system in different fields, and some studies on the immune mechanism of the Cas13 family.
- For example, the authors did not describe and discuss the recently found Cas13X and Cas13Y systems. May be other systems has been found?
Response: According to your valuable comments, we have added related discussion and description about Cas13X and Cas13Y in the new version, which mainly includes three parts. Firstly, it is about the source, function, and structure information of Cas13X and Cas13Y. Secondly, the molecular mechanism of RNA cleavage of Cas13X was added. The third part is about some application research of Cas13X.
- Section 2 The authors need to describe the current data on how the CRISPR/Cas13 system initially recognizes target RNAs, converts them to DNA, and inserts the appropriate spacers into the CRISPR loci.
Response: According to your valuable comments, we added information about the immune mechanism of the Cas13 family in Section 2.3 of this review, mainly including the process of foreign RNA recognition, transcription and recombination into the CRISPR system of Cas13a and Cas13d, and added the mechanism diagram of this process in Figure 3. However, due to the incomplete research on the immune mechanism of other Cas13 effectors, the mechanism and process of Cas13 effectors could not be comprehensively described in this review.
5.Section 2.2 Authors need to describe the structure of the recently discovered VI-X (Cas13X) and VI-Y (Cas13Y) systems (e.g. DOI: 10.1038/s41592-021-01124-4).
Response: Now, we added important information about Cas13X and Cas13Y in Section 2.2 of the revised review, mainly including their source, structure, function, and other information.
The main modification is as follows: " In 2021, Xu et al. developed a computational pipeline to search for previously uncharacterized CRISPR–Cas13 systems from metagenomic datasets. Using CRISPR arrays as search anchors, they identified 340425 putative CRISPR repeat arrays and sequentially analyzed seven Cas13 variants based on conserved stem-loop structure and BLAST alignment similarity, and finally divided them into two groups, including Cas13X (Cas13X.1, Cas13X.2) and Cas13Y (Cas13Y.1 to Cas13Y.5), which were identified from hypersaline samples (Cao et al., 2022; Xu et al., 2021). The newly identified Cas13X shares some similarity with the previously identified Cas13 family effectors, but is shorter and more compact in structure (Nakagawa et al., 2022). The crystal structure analysis revealed that Cas13X protein adopts a bilobed structure consisting of REC and NUC lobes, and crRNA DR could be anchored in REC leaves. The NUC lobe is composed of HEPN1 and HEPN2 domains, while the REC lobe is composed of Helical 1, Lid, and Helical 2 domains interleaved with each other. HEPN1 and HEPN2 regions are connected to the REC lobe by inter-domain linkers 1 (IDL1) and 2 (IDL2), respectively (Nakagawa et al., 2022). Our phylogenetic analysis revealed that Cas13bt proteins are not monophyletic relative to Cas13b proteins, suggesting that Cas13bt proteins evolved from larger ancestral Cas13b proteins through multiple deletions. The crRNA consists of the 5-nucleotide spacer (guide) segment and the 36-nucleotide DR region. The DR region comprises stem 1, an internal loop, stem 2, and a hairpin loop. The electron density was less distinct for the spacer region, suggesting its flexibility in the Cas13bt3crRNA binary complex structure (Nakagawa et al., 2022).".
- Section 4 Authors need to describe the application of the recently discovered VI-X (Cas13X) and VI-Y (Cas13Y) systems. Cas13d is not the smallest protein of type IV system anymore. The authors did not describe applications of Cas13 effectors in possible therapeutic application by the regulation of mRNA modifications, splicing and translation.
Response: We have added the application research of Cas13X and Cas13Y in the application description section of the review.
First, in antiviral applications, the content " Yang et al. applied AAV-PHP.eB capsid and GFAP promoter to drive the specific expression of Cas13X-NLS-HA-sgPtbp1 in astrocytes and showed that the PTBP1 signal, a key signal, was decreased after 2 weeks, 1 month and 2 months of intravenous treatment, respectively. The PTBP1 S signal decreases sequentially from 74.79% to 18.42%, 14.96%, and 11.98%, which indicates that astrocytes can be successfully converted into neurons (AtN) (Yang et al., 2022)." has been added.
- In line 34, remove the duplicate word "identified" in the phrase "first identified and identified a."
Response: We remove the duplicate word "identified" in the phrase "first identified and identified a." in the revised manuscript.
Minor points.
- In the introduction, the authors need to mention the cornerstone work of Martin Jinek et al. DOI: 10.1126/science.1225829, where they elegantly showed the complete mechanism of action of CRISPR/Cas9. This work has been followed by others showing that CRISPR/Cas9 can be applied to edit the human genome and other organisms. Line 45 Since the authors are talking about two articles, they need to refer to the second article DOI: 10.1126/science.1232033.
Response: Based on your valuable comments, this review further improves the introduction and adds the important cornerstone work of Jinek in this field.
- Lines 45-47. The authors need to be more specific: Cpf1(Cas12a) recognizes AT-rich PAMs, so it can target AT-rich regions where can be no GC-rich PAMs for Cas9, but Cpf1(Cas12a) has difficulty finding a target in GC-rich regions. Therefore, Cpf1(Cas12a) together with Cas9 expands the number of possible target regions.
Response: We made a more detailed description of Cas12a in this review, and proposed that Cas12a and Cas9 jointly expanded the possible target area due to the difference in the PAM region between Cas12a and Cas9.
- Line 49 The authors should be more precise in their terminology. "Genetic DNA of organisms" is an awkward term. It is better to say DNA.
Response: Thank you so much for reviewing our manuscript! We have modified "Genetic DNA of organisms" to "DNA" in the new version.
Round 2
Reviewer 1 Report
The authors sufficiently added the points
Author Response
Dear editor & reviewers,
Thank you for handling our manuscript entitled " Insights gained from RNA editing targeted by the CRISPR-Cas13 family "! We also thank the reviewers for the valuable and constructive advices! All those efforts really improve the level of our manuscript. The point-to-point responses are listed below, and the new revision is marked red. On behalf of all authors, we thank you all again!
Yours sincerely,
Desheng Pei Ph. D.
Professor of Chongqing Medical University (CQMU)
No.1 Yixueyuan Road,
Yuzhong District,
Chongqing 400016, China
Line 178 and following: "and finally divided them into two groups, including Cas13X (Cas13X.1 and 178 Cas13X.2) and Cas13Y (Cas13Y.1 to Cas13Y.5) (Cao et al., 2022; Xu et al., 2021). The newly 179 identified Cas13X shares some similarity with the previously identified Cas13 family ef-180 fectors, but possesses a shorter and more compact structure (Nakagawa et al., 2022). The 181-crystal structure analysis revealed that Cas13X protein adopts a bilobed structure..." You describe Cas13X but you indicate nothing about Cas13Y. Could you add a sentence about it?
Response: Thank you so much for the valuable comment! Nothing is known about Cas13Y due to the lack of new research results on Cas13Y. Therefore, in the revised manuscript, we added a related explanation: “However, currently, no detail structure information of Cas13Y is reported”.
Reviewer 2 Report
The authors have done a good job of greatly improving their review, making it newer, more comprehensive, and more useful to readers. I can now recommend the revised version of the review for publication in the IJMS.
Author Response

(The authors gave the same response as above.)
